

# Flash-flood forecasting in two Spanish Mediterranean catchments: a comparison of distinct hydrometeorological ensemble prediction strategies

VINCENDON Beatrice[1] and AMENGUAL Arnau[2]

[1]CNRM (Météo-France, CNRS),42 av. Coriolis, 31057 Toulouse Cedex, Toulouse, France
[2]Grup de Meteorologia, Dep. De Física, Universitat de les Illes Balears. Palma, Mallorca, Spain

*Correspondence to:* Vincendon B.(beatrice.vincendon@meteo.fr)

**Abstract.** Hydrological Ensemble Prediction Systems (HEPSs) are becoming more and more popular methods to deal with the meteorological and hydrological uncertainties that affect discharge forecasts. These uncertainties are particularly difficult to handle when dealing with Mediterranean flash-flood forecasting as many hydrological and meteorological factors take place and precipitation comes from small scale convective systems. In this work, the performances of distinct HEPS are compared
for two heavy precipitation events that affected two different semi-arid Spanish Mediterranean catchments : the cases of the 03 November 2011 on the Llobregat River in Catalonia, and the 28 September 2012 on the Guadalentín River near in Murcia. The latter case corresponds to the IOP8 of HYMEX field campaign. The uncertainty on quantitative precipitation forecasting is sampled by using two different meteorological ensemble generation strategies. First, a convection-permitting EPS, which consists in dynamically downscaling the ECMWF-EPS directly by means of the WRF model. The second EPS strategy is
based on the AROME-WMED convective-scale model. Its deterministic QPFs are perturbed based on a previous rainfall forecast error climatology and by using the probability density functions of the errors, in term of total amounts and location of the heaviest rainfalls. The population of both ensembles is of 50 members, which are used to drive the HEC-HMS and ISBA-TOP hydrological models. For each HEPS, the performance is assessed in term of the quantitative discharge forecasts. The results point out the benefits of using (i) a hydrological model when evaluating highly-variable and convective-driven
precipitation fields and (ii) an EPS to better encompass these uncertainties arising from different level of the HEPS. Issues about the optimal number of ensemble members and impact of the ensemble forecasting lead time are addressed for optimal flash-flood forecasting purposes as well.

## 1  Introduction

Flash floods are among the worst hazards worldwide (Doocy et al., 2013). These hydrometeorological episodes can result in substantial human, social and economic losses. The HYdrological cycle in the Mediterranean EXperiment (HyMeX, http://www.hymex.or





programme fosters a better understanding, quantification and modelling of precipitation and flood extremes in the Mediterranean (Drobinski et al., 2014). The Spanish Mediterranean region is affected recurrently by extreme precipitations resulting in flash-flooding, mainly during the end of the warm season. The early-autumn intrusion of upper-level cold air masses in the Western Mediterranéean region and its comparatively large sea surface temperature boost the convective available potential
energy. Besides, the prominent orography of Mediterranean Spain results in the lifting of the warm and moist sea and the subsequent generation of deep moist convection (Fig. 1).

Flash floods result from the persistence for several hours of high precipitation rates over specific hydrographic catchments. In the Spanish Mediterranean region, quasi-stationary precipitation is often a consequence of long-lived Mesoscale Convective Systems (MCSs) which remain anchored by the prominent orography (Romero et al., 2000; Ducrocq et al., 2014). In addition,
many semi-arid river basins are small-to-medium sized, highly urbanized and contain coastal steep streams. All these factors shorten even more the hydrological response times. As many small rivers are ephemeral, large and rapid flows carrying extensive quantities of debris can exacerbate extensive flood damage. Therefore, the development and evaluation of state-of-the-art hydrometeorological tools is an issue of major relevance. These tools can contribute to a better understanding and forecasting of flash floods in order to implement more reliable forecasting and warning systems over the Mediterranean Spain.

For this purpose, we have selected the 03 November 2011 and the 28 September 2012 flash floods in the Llobregat and Guadalentín basins, respectively. The Guadalentín and Llobregat are an archetype of Spanish Mediterranean catchments that are recurrently affected by flooding (for further details, see Amengual et al. (2007, 2009, 2015) ). The Guadalentín river basin is located in Murcia while the Llobregat watershed is located in Catalonia, south and north-eastern Spain respectively(Fig. 1). For both catchments, we have implemented two event-based models : (i) the semi-distributed and conceptual-based HEC-
HMS model (USACE-HEC, 2000) and; (ii) the fully-distributed and physically-based ISBA-TOP model (Bouilloud et al., 2010; Vincendon et al., 2010, 2016).

The first step of the present work is to explore the impact on the simulated flood hydrographs of these distinct hydrological model set-ups that account for different levels of variability in the rainfall fields and basin properties. This approach allow examining the impact of different model structures, physical parameterizations, complexity and modeling scale (Smith et al.,
2004). The use of distinct hydrological models allows investigating more comprehensively how impact the aforementioned modelling issues the real-time modelling of flash-flooding.

Regarding the aim of the present work, the use of two models with different structures, especially for the modelling of the soil infiltration mechanism, may result beneficial to better understand and describe the rainfall-runoff transformation processes, according to the nature of the rainfall episode which occur over the catchment in question. Specifically, we use two distinct
model structures and physical parameterizations to simulate the complex rainfall-runoff transformation of intense precipitations over semi-arid basins. In fact, the different soil infiltration and routing schemes are determining factors modulating peak discharges, timings and runoff volumes. As a matter of fact, the characteristics of the rainfall event (i.e. spatial-temporal distribution and intensity) may influence the simulated catchment's response depending on the modelled surface runoff generating mechanism (Hearman and Hinz, 2007).



Quantitative precipitation forecasts (QPFs) by short-range and high-resolution numerical weather prediction (NWP) models are a primary tool to further extend flood forecasting lead-times beyond the basin response times. Nowadays, high-resolution NWP models capture the initiation and intensification of convectively-driven rainfalls realistically, with similar spatial and temporal scales to the flash flood-prone catchments. That is, QPFs can be employed to force rainfall-runoff models without

the need of implementing additional downscaling procedures (Amengual et al., 2008; Vincendon et al., 2011). However, hydrometeorological forecasts are impacted by different types of uncertainty that degrade the final outputs. Uncertainties arise from the physical schemes used in each particular hydrological and meteorological model as well as from their initial and lateral boundary conditions (IC/LBCs). Furthermore, QPFs of small-scale convective rainfall are challenging as many factors are involved in their determination. When mesoscale convective developments are dominant, errors can become larger during

the forecasting process and produce inaccurate QPFs.

Short-range ensemble prediction systems (SREPSs) aim at forecasting the probability of extreme weather as accurately as possible. Uncertainties in the atmospheric state are most often encompassed by running NWP models with perturbed IC/LBCs (Buizza, 2003; Grimit and Mass, 2007). Next, the distribution of plausible atmospheric states –represented by SREPSs– are used to build hydrological ensemble prediction systems (HEPSs) in order to convey these external-scale uncertainties (*i.e.*

external to the hydrological models) down to the hydrological system. That is, the inclusion of independent information from a distribution of atmospheric scenarios aims at increasing the skill of HEPSs. However, the identification of the most suitable methods for generating these HEPSs and the quantification of their added value are still under investigation (Cloke et al., 2013).

The objective of the present work is to evaluate the predictive skill of several distinct HEPSs for the 04 November 2011 and the 28 September 2012 floods. To this end, we have used the AROME and WRF meteorological models to build two

SREPSs. The WRF-SREPS has been generated by dynamically downscaling the operational ECMWF-EPS (European Center for Medium Range Weather Forecasts – global Ensemble Predictions System) forecasts. The so-called AROME ensemble has been built on the perturbation of the rainfall outputs of AROME-WMED (Fourrié et al., 2015) model by the Vincendon et al. (2011) methodology. Next, we analyze the performance of these two distinct SREPS generation strategies when forcing both hydrological models through a twofold approach. First, QPFs have been evaluated by comparing them to the interpolated rain-

gauge data. Second, QDFs have been examined by using classical probabilistic objective scores. The impact of the forecasting lead time of the SREPSs as well as the size of the ensemble have been investigated.

Next, section 2 presents a short overviiew of the flsh-floods, the study areas and observational networks; section 3 provides the hydrological tools, the atmospheric models and ensemble generation strategies; results are presented in section 4.2. The last section summarizes main conclusions and provides further remarks.

## 2  Study regions, databases and flash-flood episodes

### 2.1  The Llobregat River basin and the first select episode

The Llobregat basin is the largest internal hydrographic catchment of Catalonia (Fig. 2 a). Altitudes range from above 3,000 $m$ in the Pyrenees to between 200-750 $m$ in the pre-coastal and littoral ranges. The Llobregat river basin extends covering an





area of 5,040 km$^2$, with a maximum length around 170 $km$. The Llobregat covers a wide spectrum of annual rainfall amounts depending on altitude. Precipitation is broadly beyond 1000 $mm$ where the altitude is higher than 1000 $m$ in the Pyrenees. Annual rainfall amounts are rougly of 700 $mm$ over the pre-Pyrenees, with altitudes ranging from 600 to 1000 $m$, and barely reach 500 $mm$ for lower heights. The Llobregat basin exhibits a mild and rainy cold season and a hot and dry warm season, characteristic of the Mediterranean climate. Furthermore, extreme precipitations affect the Spanish Mediterranean region every year, which represent a substantial part of the total amounts. Two dams are found in the montainous regions of the Llobregat (Fig. 2 a).

Available raw precipitation consists of 5-minute rainfall data recorded at 81 stations inside or very close to the Llobregat basin (Fig. 2 a). These pluviometric stations belong either to the Catalan Agency of Water – Automatic Hydrological Information System (ACA-SAIH) or the Spanish Agency of Meteorology (AEMET) networks. Five-minute stream flow data is recorded at four hydrometric sections in the Llobregat River. These stream-gauges are included in the ACA-SAIH network and are deployed in: (i) Súria town, with a dranaige area of 940 km$^2$ (labelled as Súria); (ii) Sant Sadurní d'Anoia city, with a drainage area of 736 km$^2$ (Sadurní); and (iii) Castellbell (3340 km$^2$) and (iv) Sant Joan Despí (4915 km$^2$) towns (labelled as Castellbell and Despí, respectively). A more detailed description of the Llobregat river basin and databases can be found in Amengual et al. (2007).

The first selected event occured in autumn 2011, which has been loaded with heavy precipitation events (HPE) in the north-western Mediterranean (Hally et al., 2013; Silvestro et al., 2012; Rebora et al., 2013). The northern part of Catalonia was particularly affected at the beginning of November, 2011. On November 2nd, the large-scale situation was dominated by a deep, cold upper-level trough approaching from the North Atlantic Ocean while a southeasterly low-level flow was strengthening over the Catalonia region. Those atmospheric conditions favoured convection and heavy rainfall occured. A maximum amount of daily precipitation of 202.9 $mm$ was recorded on 3 November over Catalonia and slightly above 150 $mm$ over the Llobregat River basin. Most of the rivers of Catalonia were flooded even if minor damage was produced in Catalonia (Llasat et al., 2014).

## 2.2 The Guadalentín River basin and the second selected episode

The Segura is one of the most important Spanish rivers running into the Mediterranean Sea. This catchment spans over an area of 18,208 km$^2$ and the maximum length of the Segura river is about 325 $km$. The Guadalentín is the main affluent of the Segura River. The river basin extends over a region of 3343.1 km$^2$ and the river length is close to 121 $km$ (Fig. 2 b). The Guadalentín river basin comprises altitudes ranging from above 2,000 m (in the Baetic system) to 1,200 m in the Murcian pre-litoral depression, and to barely 110 m in its mouth to the Segura. The Guadalentín basin is found in a particularly arid region of Mediterranean Spain, owing to its particular settlement. This Baetic range shelters this area from the frequent passage of cold fronts coming from the Atlantic in the wet season and bringing copious precipitations to other Spanish regions. Thus, precipitation in the catchment relies on the cyclogenesis of Mediterranean systems and the subsequent impinging of humid low level flow coming from south west. But these disturbances are sparse in time and small in space. Rainfall amounts are height-dependent and range from 500 to 300 $mm$.




108 automatic rain-gauges are located within the Confederación Hidrológica del Segura (CHS) boundaries (Fig. 2 b). Raw precipitation accumulations are recorded with a time frequency of 5 minutes. These automatic stations belong either to the Confederación Hidrológica del Segura – Automatic Hydrological Information System (CHS-SAIH) or to AEMET networks. Almost 40 of these 108 stations are deployed over the Guadalentín River watershed or in the vicinity. Stream-flow is recorded at three hydrometric sections along the Guadalentín River: in Lorca and Paretón de Totana cities (labelled as Lorca and Paretón, respectively) and in Salabosque -outskirts of Murcia city-. Their respective drainage areas are of: 1827.1 km$^2$, 2384.7 km$^2$ and 3170.4 km$^2$ (Fig. 2 b). Runoff is registered in the CHS-SAIH network with a temporal frequency of 5 minutes as well.

The hydraulic section of the CHS is well aware of the short recurrence of catastrophic flash-floods affecting the whole region. Therefore, numerous structural measures have been implemented over the Segura River basin along the years. Specifically, four reservoirs are located within the Guadalentín. Furthermore, a channel was constructed downstream of Paretón so as to link directly the river with the Mediterranean Sea (Fig. 2 b). This channel was designed specially so as to prevent hazardous flooding impacting Murcia city. Therefore, this channel partially diverts large runoff discharges from the Guadalentín River into the Mediterranean Sea. Further details on the Guadalentín and databases can be found in Amengual et al. (2015).

The second study case is a classical Spanish Mediterranean HPE that affected Andalusia, Murcia and later on Valencia and Catalonia (even if less intense there) from 27 to 29 September 2012. This case has been documented during the HyMex campaign within an Intensive Observing Period, IOP8 (Amengual et al., 2015). About ten casualties have been deplored and damage has been estimated to more than 120 Meuros. On 27 September 2012, at upper-level, the synoptic atmospheric situation was controled by a cut-off low centered on the South-West of the Iberian Peninsula moving slowly north-eastward and a low-level depression centered upon inner Andalusia. The upwards forcing associated with the north-east flank of the cut-off low favoured the trigerring of deep convection and the development of a low-level convergence zone. Those ingredients reinforce convection and heavy precipitation across Murcia, Valencia, the Balearics and Catalonia, that was affected on 30 September 2012. The amounts of precipitation locally reached 214 $mm$ in 24 hours in Andalusia and 240 mm in Murcia. They caused the flash-flooding around Murcia, especially the one of the Guadalentín River (Ducrocq et al., 2014; Amengual et al., 2015).

# 3 Models and methodology

## 3.1 Meteorological models

### 3.1.1 The WRF model

The Weather and Research Forecasting (WRF) model version 3.4 (Skamarock, 2008) have been implemented with a single computational domain of 767 x 575 grid-points that is centered in the Western Mediterranean and spans the entire Mediterranean Spanish coast (Fig. 1). We have used a horizontal resolution of 2.5 km, 50 vertical levels and an integration time-step of 12 s. This model set-up allows resolving explicitly deep moist convective systems with relevant entity (Weisman and Klemp, 1997; Bryan et al., 2003; Roberts and Lean, 2008; Zheng et al., 2016).





The operational European Center for Medium Range Weather Forecasts – global Ensemble Predictions System (ECMWF-EPS) aims at coping with uncertainty on the initial atmospheric conditions, taking all the observed and modelled information available into account. In particular, the global ECMWF-EPS consist of 50 integrations with a spatial resolution of $\sim$ 20 km, after applying singular vector perturbation to an initial forecast (Palmer et al., 1998; Leutbecher, 2005; Hoskins and Coutinho, 2005). By a dynamical downscaling of the global ECMWF-EPS, we rely on the sampling of the IC/LBCs uncertainties provided by the system, but we simulate more precisely the small-scale interactions so as to enhance the high-resolution quantitative precipitation forecasts (Branković et al., 2008). Lateral boundary fields are updated every 3-h.

Physical parameterizations are identical across all WRF ensemble members and involve: the WRF single-moment 6-class microphysical scheme with graupel (WSM6; Hong and Lim (2006)); the 1.5-order Mellor-Yamada-Janjić boundary layer scheme (MYJ; Janjić (1994)); the Dudhia short-wave scheme (Dudhia, 1989); the RRTM long-wave scheme (Mlawer et al., 1997); the unified NOAH land surface model (Tewari et al., 2004); and the eta similarity surface layer model (Janjić, 1994). Atmospheric simulations span over two consecutive 48-h periods for the Llobregat event: 02-04 November and 03-05 November 2011 00 UTC, respectively. For the Guadalentín episode, NWP simulations span over two consecutive 48-h periods: 27-29 September and 28-30 September 2012 00 UTC, respectively. Next, hourly QPFs outputs are used to force the hydrological models. Note that the model set-up matches the WRF operational configuration employed at the University of the Balearic Islands ($http://meteo.uib.es/wrf$).

### 3.1.2 The AROME-WMED model

AROME-WMED is a research convection-permitting model dedicated to HYMEX (Fourrié et al., 2015). It is based on AROME-France that is the operationnal non-hydrostatic model of Météo-France (Seity et al., 2011). The models run at a 2.5-km horizontal resolution. It has 60 vertical levels that range from 10 m above ground to 1 hPa. The deep convection is thus explicitly resolved. Its microphysical parameterization is a one-moment 5-class scheme (Pinty and Jabouille, 1998; Caniaux et al., 1994) with rain, snow, graupel, cloud liquid water and cloud ice. In the boundary layer, the vertical turbulent transport follows by two schemes: the parameterization of (Cuxart et al., 2000) for prognostic turbulent kinetic energy for an eddy diffusivity part or the mass flux scheme of (Pergaud et al., 2009) for dry thermal and shallow convection. The surface scheme is SURFEX (Masson et al., 2013) and the surface boundary layer is SLB (Masson and Seity, 2009). AROME-WMED covers a domain that encompasses most of the Western Mediteranean Sea (34N-11W,48N-20E; see Fourrié et al. (2015) to vizualise it). Its lateral boundary conditions come from forecasts of the French operational global model ARPEGE (Courtier et al., 1991). During HYMEX SOP1, a daily 48-h AROME-WMED forecast was run in real time, starting at 00:00 UTC every day from september to december 2012. A 3D-var data assimilation scheme is used to produce the initial atmospheric state. Compared to AROME-France analysis, more satellite and surface observations and some experimental measurements such as low-layer ballons data and additional radiosoundings were included in the data assimilation process. The overall quality of the AROME-WMED model proved to be as good as the AROME-FRANCE one, especially QPFs. All details can be found in Fourrié et al. (2015).



Vincendon et al. (2011) used the object-oriented verification method called SAL (Wernli et al., 2008) to evaluate AROME QPF, in terms of location and amplitude errors. The references used were the quantitative rainfall estimates (QPE) provided by radar data (see Tabary, 2007;Tabary et al., 2007). No systematic biases in the magnitude of the rainfalls were found and location errors remained lower than 50 $km$ in the 80% of the cases. So, the high-resolution process-based model trajectory can

be used to feed an hydrological model subject to the consideration of the uncertainty that affects QPF.

A classical way to take the uncertainty of atmospheric forecasts into account is to use a meteorological EPS. A prototype of a meteorological ensemble at convective-scale based on AROME model has been developed (Vié et al., 2011; Nuissier et al., 2012; Bouttier et al., 2012). This system is called AROME-EPS. Unfortunatelly, the AROME-EPS domain does not encompass the Murcia area. So, it has not been used in this work and another approach has been adopted.

Vincendon et al. (2011) developed the so-called "Perturbation method" so as to create a set of QPF scenarios on the basis of the AROME-France deterministic forecast that contains valuable information. Primarly, an object-oriented climatology of forecasts errors was established besides the SAL evaluation cited in the former section. First rainy objects were defined as corresponding to connected grid cells with more than 2 $mm$ an hour. The same is performed for convective objects, which were connected grid cells with more than $100mm$ an hour. Some probability density functions (PDF) of errors in term of

amounts of rain and location of those objects were computed. Then a perturbation tunned from those PDFs was introduced into the deterministic forecast. The perturbation method follows the following steps:

- Shifting of the rainy objects in accordance with the PDF of the errors in location of the AROME deterministic forecasts.

- Change of the rainfall intensity inside the rainy objects in accordance with the PDF of the errors in amplitude of the rainy objects.

- Change of the convective objects within each rainy object that are set more or less peaked/flat in accordance with the PDF of the amplitude errors of the convective objects.

Next, this method was applied to the AROME-WMED model forecasts. This is possible owing to the very close statistics obtained with AROME-France and AROME-WMED in QPF evaluations.

This system is not a meteorological EPS but it allows to obtain several members for AROME forecasts. Consequently, the

system will be called AROME ensemble in the following in order to simplify the writing and understanding. In this work, 48-h AROME-WMED forecasts started on 02 and 03 November 2011 at 00 UTC for the Llobregat study case, and 27 and 28 September 2012 at 00 UTC for the Guadalentín study case.

### 3.2 Hydrological models

#### 3.2.1 The HEC-HMS model

HEC-HMS has been used as semi-distributed, conceptual- and event-based configuration for both river basins (USACE-HEC, 2000). A single hyetograph is used to drive the hydrological model for each subbasin. First, the hourly rainfall spatial accumulations are obtained from the 40 rain-gauges available. Spatial distributions are obtained by the kriging method. The semi-variogram has been matched after applysing a linear model. Next, we have computed the time series of the hourly rainfall



amounts for each individual subwatershed as the area-average of the spatially gridded precipitation within the corresponding subbasin. Note that the same procedure is used to drive the hydrological model with the QPFs, but we have used NWP model grid points instead of the observed precipitation.

Runoff is computed according to the curve number methodology (CN; USDA, 1986). CNs depends non-linearly on a wide range of facets: accumulated precipitation, lithology, land uses and soil's antecedent moisture condition (AMC; (Chow and W., 1988)). CNs were obtained after experimental field campaigns with AMC II for both river basins. The dimensionless unit hydrograph (UH) provided by the American Soil Conservation Service has been applied so as to transform the effective rainfall into overland flow for every sub-basin. This conceptual scheme links the runoff maximum with the time-to-peak by accounting for the sub-basin area and a conversion constant (USACE-HEC, 2000). The Muskingum method has been implemented so as to propagate the flood hydrograph (Chow and W., 1988). Therefore, the model set-up employs spatially uniform conceptual model parameters for each sub-basin and for the dynamical formulation as well.

The dams have been modelled by using the following information provided by the CHS and ACA hydraulic divisions: the storage capacity, maximum outflow and elevation, and initial water level. The reservoirs have been simulated by using their elevation-storage-outflow relationships (USACE-HEC, 2000). Finally, we have implemented a diversion element to account for the diversion of the flood volumes towards the Mediterranean Sea in the Guadalentín river. Note that data on diverted flows have been provided by the CHS.

Model calibration focused on peak discharges, timings and runoff volumes, which are strongly dependent on infiltration and routing processes. In the semi-arid Mediterranean Spain, sparse vegetation together with torrential convective precipitations, that easily exceed the high initial soil infiltration capacity, favour fast Hortonian flows and rapid flow velocities in the river streams (Belmonte and Beltrán, 2001). Consequently, we calibrated the following model parameters: SCS-CNs, times of concentrations and flood wave celerities. Note that SCS-CNs encompass effectively the initial soil moisture conditions. The calibration procedure was carried out by combining a manual and an automatic approach. The automatic procedure uses as objective function the root mean square error weighted according to the peak; and as search algorithm, the univariate-gradient method (USACE-HEC, 2000). A complete description of the HEC-HMS set-ups and the calibration and validation tasks for both catchments can be found in (Amengual et al., 2007, 2009, 2015).

All the hydrologic simulations comprise a 96-h period. For the Llobregat flooding, HEC-HMS has been run from 02 to 06 November 2011 00 UTC, with a 10 minute time-step. For the Guadalentín flash flood, model simulations span from 27 September to 01 October 2012 00 UTC, with a 5 minute time-step. These simulation periods encompass the whole flash-flood episodes. Note that the hydrological model applies a linear interpolation to convert hourly rainfall amounts to the model time-step.

### 3.2.2 The ISBA-TOP model

The hydrological model ISBA-TOP (Bouilloud et al., 2010) is dedicated to Mediterranean catchments simulations. ISBA-TOP fully couples the land surface model ISBA (Interaction Surface Biosphere Atmosphere, Noilhan and Planton, 1989) and a version of TOPMODEL (Beven and Kirkby, 1979) that has been adapted to the Mediterranean areas (Pellarin et al., 2002).





This coupling consists of introducing a lateral distribution of soil water following the TOPMODEL concept into ISBA. ISBA deals with the soil-atmosphere exchanges : water and energy budgets are managed over a rectangular domain described by 1 $km^2$ grid cells. The hydrological processes are simulated over soil vertical columns. The catchments are described by small grid cells (called pixels) according to a DTM. The ISBA soil moisture over a grid cell allows determining the water-storage deficit on the corresponding catchment pixels as well as the hill slope recharge. TOPMODEL equations allow computing the lateral water transfers within the catchment using the topographical information and rainfall spatial distribution. The pixels water-storage deficit are thus updated and are transformed back into ISBA soil moisture. Pixels with null deficits define the saturated contributive areas.

ISBA computes soil water flows from those new saturated areas and soil moisture fields. Runoff over saturated areas (Dunne, 1978) occurs when water excess concerns the contributive areas that are diagnosed by the TOPMODEL approach. Hortonian runoff that occurs when rainfall intensity exceeds the infiltration capacity (Horton, 1933) is estimated on the the non-saturated fraction of the grid. ISBA also computes gravitational drainage at the bottom of the deepest soil layer. The drainage flow computed for an ISBA grid cell is distributed over all the corresponding catchment pixels, whereas total runoff is assumed to occur on the saturated catchment pixels only. The DTM informs about the geomorphology of the catchment, including the distance between each hill slope pixel and the river. The water flows are routed up to the river so as to compute total discharges at each river pixel through a geomorphological method. Artinyan et al. (2016) link the river discharge variable and the water velocity in the river.

A full description of the coupling principle is available in Bouilloud et al. (2010). This ISBA-TOP original version has recently been improved so as to obtain satisfactory results without a calibration of the parameters. They are defined through pedotransfert functions (Vincendon et al., 2016). Reservoirs and diversion elements have been considered on the same basis as described in section 3.2.1.

In this work, ISBA-TOP has been run with an hourly time step. Hourly rainfall amounts collected by rain gauges and spatially distributed (through a linear horizontal interpolation method) are used to drive ISBA-TOP. Other parameters (2m-air temperature, 2m-air humidity,10m-wind speed and direction,...) are set constant. AROME-WMED analyses provide ISBA-TOP initial soil water contents and temperatures. ISBA-TOP simulations start systematically 2 days before the beginning of the studied period, so on 01/11/2011 at 00UTC and on 26/09/2012 at 00UTC to ensure soil moisture balance at the beginning of the rainfall event.

### 3.3 Designed HEPS

Several HEPS have been built by coupling HEC-HMS and ISBA-TOP hydrological models with both WRF and AROME-WMED SREPSs. Recall that the SREPSs have been run for two 48-h different periods for each case study: on 02/11/2011 00 UTC and 03/11/2011 00 UTC for the Llobregat flood, and on 27/09/2012 00 UTC and 28/09/2012 00 UTC for the Guadalentín flooding. Each SREPS has 50 members plus the unperturbed simulation. The resulting HEPSs have been labelled as: AROME-ISBA, AROME-HMS, WRF-ISBA and WRF-HMS, respectively, and have been run for the aforementioned periods. In brief, we have four experimental set-ups encompassing the 03/11/2011 and 28/09/2012 floods: RUN02 (starting on 02/11/2011 00





UTC), RUN03 (starting on 03/11/2011 00 UTC), RUN27 (starting on 27/09/2012 00 UTC) and RUN28 (starting on 28/09/2012 00 UTC). For each experimental set-up we have two different 48-h EPSs and four subsequent HEPS. Several experiments have been designed for evaluation purposes. They are liste in Table 1 together with their starting time and experimental name. Summarizing the different experiments, we have distinct types of data that have been used to drive the hydrological models:

– A reference simulation consists of using rainfall collected by raingauges and spatially interpolated by kriging. These rainfall data will be called QPE, Quantitative Precipitation Estimates. The discharge simulations obtained with both hydrological models driven by measured rainfall are annotated $REF2011$ for the November 2011 case and $REF2012$ for the HYMEX IOP8 case (see Table 1).

   – The hydrological models driven by deterministic QPF (Quantitative Precipitation Forecasts) allow to obtain deterministic
QDF (Quantitative Discharge Forecasts). AROME produces deterministic QPF originally (as described in section 3.1.2 ) while, for the WRF model, the unperturbed member of ECMWF-EPS dowscaled by WRF is used as control and will be denoted as "deterministic" as well. For each event, two simulations of deterministic forecasts have been produced at two distinct starting times.

   – Ensemble rainfall scenarios provided by WRF and AROME SREPSs allow to obtain ensembles of QDFs. The same
experiments than those deterministic are produced (two starting times for each study case).

   Finally, more than 2400 discharge time series have been obtained.

## 4   Results

### 4.1   Hydrological models uncertainty

The different experiments have been assessed at different outlets for each river basin, where the stream mesurements are
quality-controled by the ACA and CHS hydraulic divisions:

   – Despí, Castellbell and Sadurní for the Llobregat River and

   – Lorca and Paretón outlets for the Guadalentín River.

Figure 3 presents the cumulated Nash efficiency frequencies for hourly discharges at all outlets simulated within $REF2011$ and
$REF2012$ experiments with HEC-HMS model (in blue) and ISBA-TOP (in orange). These empirical cumulative distributions are built following the method of Le Lay and Saulnier (2007). The Nash efficiencies computed for each simulation are ranked from the smallest to the largest. The probability of a value being less than the $i_{th}$ smallest Nash (given by $(i - 0.5)/n$, with $n$ total number of data) gives the cumulative frequency. On the graph, the more the distribution is shifted to the right, the better is the skill of the model. This representation allows to characterise the overall accuracy of a model from a regional point of
view. Better results are obtained with HEC-HMS model with more frequent high Nash efficiencies. This is confirmed by the



predictive uncertainty (see Fig. 4) that is better with HEC-HMS model. Note that HEC-HMS was previously calibrated and validated by using observed streamflow and precipitation data for both river basins Amengual et al. (2007, 2009, 2015), while ISBA-TOP has been run without any previous calibration task as described by Vincendon et al. (2016). Thus, ISBA-TOP has been initialized by using an spatial estimation of the initial soil moisture. It is worth noting that the simpler semi-distributed

and conceptual HEC-HMS has shown a relatively high skill when simulating the observed floods for the Llobregat and the Guadalentín basins. However, without the preceding tasks of calibration and verification, the runoff model had exhibit a rather deficient performance for both episodes. That is, it does not appear to be suitable implementing simple conceptual models for flash-flood forecasting purposes over the semi-arid Spanish Mediterranean basins without a prior adjustment of the hydrological parameters. The initially dry soils and high infiltration capacities of these semi-arid catchments enhance the nonlinear response

of runoff to intense precipitation and large rainfall amounts. In addition, heterogeneities also arise in the hydraulics of these basins' response to flash floods (Amengual et al., 2017). Not previously calibrating and verifying such simple conceptual hydrological models would hamper to cope with these high nonlinearities.

An illustration of the raingauge driven simulated hourly discharge time series is given by Figures 5 and 6, that show some examples of deterministic as well as ensemble forecasts. For the November 2011 case, the flow peaks obtained with ISBA-TOP

driven by QPE are underestimated compared to the observations at Despí and Castellbell (see Fig. 5a) and overestimated at Sadurni. Two peaks of flow are obtained with both hydrological models, a moderate time lag is obtained with HEC-HMS that simulates a peak of the good order of magnitude (Fig. 5b) except for Sadurní for which an overestimation is obtained. That is, ISBA-TOP presents more inaccuracies when modelling the surface flow of the first wave of intense rains, probably as consequence of estimating the initial conditions of the soil drier than the actual. The simulations of hourly discharges by HEC-

HMS driven by QPE are quite close to the observed discharges for the September 2012 case (Fig. 6b), slightly underestimated with ISBA-TOP (Fig. 6a). Naturally, the distinct physical schemes of both hydrological models lead to distinct hydrological responses yielding useful information from both configurations. The calibrated HEC-HMS model simulates more properly the peak flow amplitude but the timing seems better reproduced by the uncalibrated ISBA-TOP coupled system as it simulates more accurately the dynamic processes of this flooding. This tends to show that a multi-model approach can be really informative.

## 4.2 Skill of the designed HEPS

### 4.2.1 QPF evaluation

Catchment-averaged precipitation amounts have been computed for 24-h periods for each experiments of Table 1 for the distinct rainfall sources : QPE based on raingauge measurements, deterministic QPFs from AROME and WRF models and ensemble forecasts. Figures 7 and 8 show the results for the cases of November 2011 and September 2012, respectively. The Llobregat

catchment-averaged precipitation amounts from 03/11/2012 at 00UTC to 04/11/2012 at 00UTC is over-/underestimated by AROME/WRF deterministic QPFs QPF for both $RUN02$ and $RUN03$ (see Fig. 7). On the contrary, all the ensemble strategies succeed in encompassing the raingauges value either in the interquartile range (for the AROME ensemble) or in the "highest





scenarii" for the WRF ensemble as figure 7 shows. Note also the generalized higher spreads for AROME than for WRF ensembles.

Deterministic QPF from both models and both starting times underestimate the 24h-accumulated rainfall averages on the Guadalentín catchment (see Fig. 8). The scenarii of rainfall forecasted by the ensemble do not suceed in correcting this ten-

dency, except for the $RUN28$ of AROME ensemble. These facts illustrate the arduous task of properly simulating extreme precipitations over medium-sized basins in terms of location, timing and rainfall amounts. Recall that small-scale convective systems are highly sensitive to different NWP model configurations, physical schemes and IC/LBCs.

As far as rainfall spatial distribution is concerned, AROME deterministic QPF overestimate the highest rainfall for $RUN02$

and $RUN03$ experiments and locates them too north-west compared to the observation. The location is better for WRF deterministic QPF but the rainfall amounts are underestimated(not shown). The behaviour of the ensembles is illustrated by figure 9. AROME ensemble still leads to overestimated the spatial extension of the highest precipitation whereas WRF ensemble still mislocate them. So the biases of the deterministic NWP models are still present in the ensembles. AROME $RUN27$ and WRF $RUN27$ deterministic as well as ensemble runs do not succed in simulating neither good rainfall amounts nor good spatial

distribution.(see Fig. 10). WRF $RUN28$ forecasts a good location for the heaviest rains but the amounts are far too weak (except for the highest percentiles of the ensemble). The contrary is obtained with AROME $RUN28$ whose QPF reach the good totals but with a mislocation. AROME $RUN28$ has a clear tendency to extend more west the rainfall. That is, WRF fails to simulate realistically the extreme rainfall values produced by the convective-scale systems anchored by the complex orography, while AROME fails to locate them realistically. It is worth noting that the WRF ensemble does not account for those uncer-

tainties associated to the physical parameterization diversity. Some previous studies have shown the benefits of using multiple physics ensemble strategies to further reduce biases when forecasting HPEs in the Mediterranean Spain (Tapiador et al., 2012; Amengual et al., 2017).

### 4.2.2 QDF evaluation

The raingauge driven discharges remain uncertain as shown in section 4.1 due to both hydrological modelling and QPE uncer-

tainties. Therefore to assess the QPF performance without being affected by hydrological modelling uncertainty, the reference will be the raingauge driven discharge simulations rather than the observed flows, that is to say $REF2011$ and $REF2012$ experiments. As aforementioned, figures 5 and 6 show some examples of hourly discharge time series. Those examples illustrate that the HEPS approach improves the forecast compared to the deterministic experiments for both extreme floods. Indeed, the inter-quartile envelopes better encompass the raingauge driven discharges (blue lines) than the deterministic forecats (green

line). This conclusion is the same than what was concluded for rainfall. A focus on the $RUN03$ experiment is interesting. The observed rainfall is well approximated by the AROME ensemble 25%-percentile or by the WRF ensemble 75%-percentile (see Fig. 7). But as far as the discharges are concerned (see figure 5), the peak discharge simulation is close to the 75%-percentile for the AROME ensemble driven HEPS and is out of the interquartile range for the WRF ensemble driven HEPS. This fact highlights the highly nonlinear nature of the rainfall-runoff transformation in semi-arid basins.





The cumulative distribution functions (CDFs) for the peak flows at Castellbell and Paretón are plotted for the previously comented examples (Figs. 11 and 12). This additional information further complements the benefits of accounting for HEPSs when coping with flash floods. Peak discharge exceedance probabilities $[P(Q \geq q)]$ of the observed flows quantify the likelihood of forecasting these extremes. $[P(Q \geq Qobs)]$ are 0.36, and 0.20 for the AROME-ISBA HEPSs at Castellbell and Paretón,

respectively. Similarly, peak discharge exceedance probabilities are 0.18 and 0.16 for the WRF-HEC HEPSs (Table 3 ). As reference, AEMET issues a warning when the probability of occurrence of extreme weather exceeds 0.20. In addition, almost all HEPSs issue unequivocal probabilities of surpassing the different discharge return periods ($Qp_T$), spaning from 0.34 to 0.98 depending on the hydrometric section (Table 3 ; for further information about the values of the $Qp_T$s, see Amengual et al. (2009, 2015).

Even if the observed peak discharge exceedance probabilities range from low to moderate for both HEPSs, the AROME-ISBA HEPSs would have triggered emergency procedures before both episodes according to the AEMET criterion. Furthermore, the distinct hydrological ensemble strategies would prove useful for conveying proper information to civil protection and emergency decision-makers before both floods as $[P(Q \geq q)s]$ are all well above 0.2. Note that the different discharge return periods quantify the risk of facing hazardous floodings.

The hourly discharges ensemble forecasts skills have been assessed computing objective scores on the whole data sample. Specifically, $RPSS$, $RMSE$, $\sigma$ and $\frac{\sigma}{RMSE}$ have been computed by using all the outlets and both cases in order to increase the statistical significance. See, for instance, Vincendon et al. (2011) for the scores formulation. Recall that the $RPSS$ assesses the benefit on the studied ensembles compared with the deterministic version of the same model. $RPSS$ score greater than zero means better skill for the ensemble than for the reference. The $RMSE$ provides a comparision with observed discharge

data and $\sigma$ informs about the spread of the HEPSs. For an informative ensemble, $RMSE$ has weak values and $\sigma$ has the same order of magnitude as $RMSE$. So ratios $\frac{\sigma}{RMSE}$ lower than one indicate a lack of spread.

Table 2 points out positive $RPSS$ values for all the ensembles that confirms a benefit of driving the hydrological models with the ensembles rather than with the deterministic QPFs. Nevertheless, the $RPSS$ is lower using WRF meteorological ensemble. The $RMSE$ values are not significantly different (less than 10% differences) contrary to values of $\sigma$ that are very

far one from the others. The spread is higher using AROME than using WRF and higher with HEC-HMS model than with ISBA-TOP. The ratio $\frac{\sigma}{RMSE}$ is however always lower than one showing a lack of spread except when HEC-HMS is driven with AROME ensemble for which the ratio is very close to 1. The lesser spread of the WRF than the AROME ensemble can be attributed to the fact that the synoptic and large mesoscale dynamical and thermodynamical environment is sufficiently accurate in the ECMWF unperturbed member. Therefore, less additional information is conveyed by the perturbed IC/LBC

ensemble members. On the other hand, the AROME ensemble is founded on a climatological model error database, accounting not just for inaccuracies in the IC/LBCs, but also in the model formulation. In this case, it would be advisable to also account for inaccuracies in the WRF model physical schemes when using a perturbed IC/LBC ensemble in order to further span the EPS spread. The different spreads in the HEC-HMS and ISBA-TOP models can be attributed to the distinct model infiltration schemes, when initiating runoff as a based-threshold process.



### 4.2.3 Impact of the forecasting lead time

In the following, the study cases are considered separately. The Brier scores computed for the QDFs starting at different times are prensented for November 2011 (Fig. 13) and September 2012 cases (Fig. 14). To increase the statistical robustness, a boot strap with 1000 repetitions has been applied. It appears that the forecasting lead-time of the atmospheric EPSs has a

strong impact on the flash-flood forecasts performances. Figure 13 shows that, for the November 2011 case, except for the HEPS where HEC-HMS is driven by AROME, the lower Brier scores are obtain with $RUN03$ rather than $RUN02$. For the September 2012 case, $RUN28$ leads to better results that $RUN27$ for the four HEPS (see Fig. 14). This tends to show that better results are obtained using the latest meteorological forecasts. This can be expected for the WRF ensemble since it is built by perturbing the initial conditions. So, the closer lead-times to the episode are, the more accurate is the representation of the

initial state of the atmospheric conditions that result in the flash-floods. With AROME, the result is more varied. Indeed this AROME ensemble comes from a perturbation of the forecasted rainfall, so it depends more on the deterministic scenario. The AROME deterministic simulations also rely on the initial conditions. So, in theory, when later the simulations start, the more accurate are the IC/LBCs to those producing the flash-floods. But closely lead-times to the episode date does not guarantees to have better simulations than starting before especially when dealing with extremes. This has to be confirmed on more a wider

climatology, but it would point out to a possible improvement of the forecasting and warning schemes before hazardous floods in order to better establish the confidence levels from an operational perspective.

### 4.2.4 Impact of the number of ensemble members

The computational cost of a HEPS with 50 members is very high for operational purposes. So, another issue of the maximum interest is to examine the ensemble forecasting skill in terms of the ensemble size. That is, are so large-sized ensembles really

necessary or smaller size ensembles have similar forecasting skill? To investigate this issue, a sub-sample of ensemble members is selected randomly among the whole population for each ensemble experiment. How is the performance affected in terms of QDF is assessed by computing the same statistical scores (i.e. $RPSS$, $RMSE$, $\sigma$, $\frac{\sigma}{RMSE}$) for the different sub-sets of HEPSs. The size of the additional ensembles range from 10 to 50 members. Fig. 15 shows the scores depending on the number of members for the September 2012 case study. Reducing the number of members leads to a deterioration of the scores but this

relationship is not linear and depends on the considered HEPS. It seems that the best compromise in terms of RPSS and spread (i. e. $\frac{\sigma}{RMSE}$ for instance) is obtained around 20 or 25 members. The same kind of results is obtained with the November 2011 case alone or with both events considered together (figs. not shown). However, the impact of the number of ensemble members is not really clear and it is difficult to point out an optimal number of members with our results. A cause might be the low number of study cases in our sample. Moreover, the selected study cases are both significant rainy events that do not cover

all the probability density function of possible rains. This might explain also that our results in term of number of ensemble members are not so clear.



## 5 Conclusions

In line with the major scientific goal of the HyMeX program of improving flood early warning procedures and mitigation measures, several distinct HEPSs have been built to forecast two flash-floods events that have occured over semi-arid Spanish Mediterranean watersheds. These HEPSs make use of two different meteorological ensembles at convective scale and two hydrological modeling systems. The performances of the hydrological models driven by observed rainfall data have been assessed first. Next, the skill of the HEPS are studied for the two flash-flood events. The aim was two-folded: to analyze the forecasting skill of several distinct HEPSs as well as to investigate the impact of the forecast lead time and of the number of ensemble members. The main conclusions of this work are :

– It has been found that the calibrated HEC-HMS hydrological model allow to better simulate the discharge peaks but the uncalibrated ISBA-TOP model better reproduces the flood dynamics.

– The use of an ensemble approach rather than a deterministic aproximation improves clearly the forecasts of both extreme rainfall episodes on the watersheds and discharges at the different hydrometric sections. This is clear whatever the case, catchment and HEPS.

– The AROME ensemble shows a higher spread than the WRF ensemble. This result tends to show that it would be useful to take the physical scheme uncertainty into account when building the WRF ensemble.

– The results of the HEPS intercomparison vary depending on the case and catchment. But it is interesting to see that the conclusions made on the catchment-averaged rainfall forecasts can be different from the ones on discharge forecast owing to the strong non linearities in the rainfall-runoff transformation. This fact shows clearly the added value of assessing ensembles through an hydrological point of view.

– As far as lead-time is concerned, a better skill is obtained for ensembles with shorter forecasting lead times, at least in the sample considered in our study.

– Regarding the ensemble size, the ensembles with 50 members obtain the best objective scores, but considering more affordable computationall ensemble sizes (around 20 or 25 members) does not deteriorate the ensemble performance significativaly

Although two case studies does not allow to reach general conclusions about the predictability of this kind of hydrometeorological hazards or about the optimal hydrometeorological forecasting strategy in an operational framework, it points out important aspects to take into account in future statistical studies. Note that the 28 September 2012 episode is a prototype of long-lasting mesoscale convective systems that are responsible for the most hazardous flash floods over the Western Mediterranean. Finally, this study aims at assessing HEPSs that combine multi-model and ensemble approaches. This kind of hydrometeorological approach will develop in the future. Integrative studies with integrated chains considering meteorology, hydrology but also hydraulics and impacts are promising approaches for a future transference to operations and breakthrough improvements in risk management strategies.



*Competing interests.* No competing interests are present

*Acknowledgements.* This work is a contribution to the HyMeX programme. The Agència Catalana de l'Aigua (ACA) and the Confederación Hidrográfica del Segura (CHS) are acknowledged for providing data. The Spanish Agency of Meteorology (AEMET) is also acknowledged for providing data from the automatic weather stations. This work has been sponsored by the CGL2014- 52199-R and PCIN-2015-221 Spanish projects, which are partially supported by FEDER funds.





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



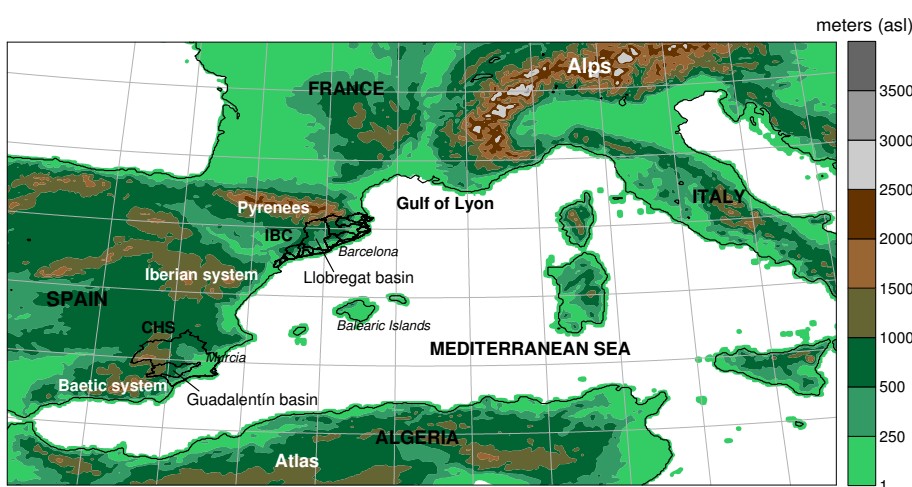

**Figure 1.** Configuration of the computational domain used for the WRF numerical simulations. Main geographical features mentioned in the text are shown. The thick continuous lines show the regions where the Llobregat and Guadalentín river basins are located in the Internal Basins of Catalonia (IBC) and the Confederación Hydrográfica del Segura (CHS), respectively.





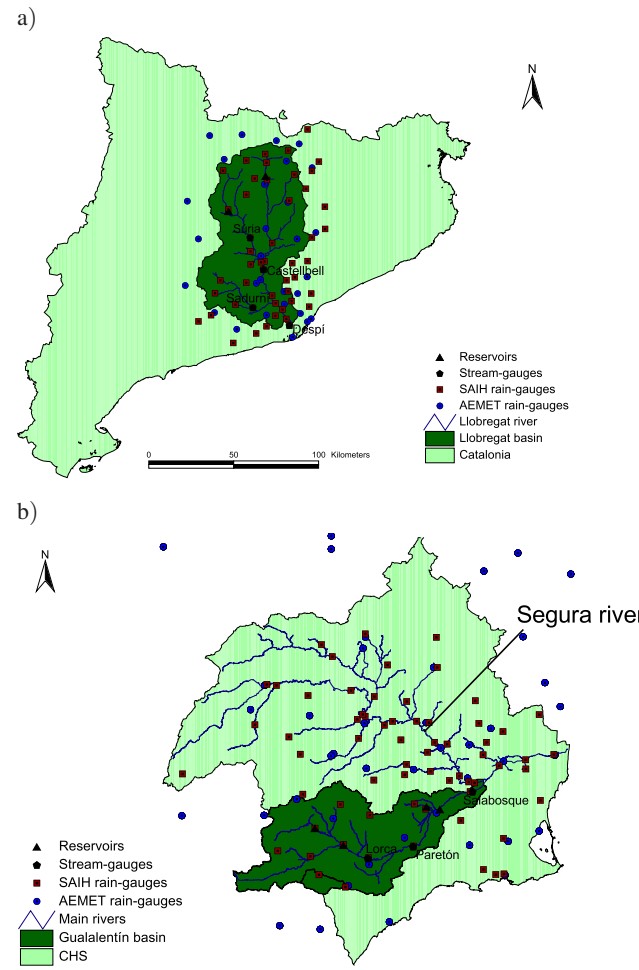

**Figure 2.** (a) Spatial distribution of the rain-gauges for the Llobregat catchment. It includes a total of 81 automatic rainfall stations. The Llobregat river basin is highlighted in shaded dark green. (b) Spatial distribution of the rain-gauges for the CHS. It includes a total of 108 automatic rainfall stations distributed over an area of $18.208\ km^2$. The Guadalentín river basin is highlighted in shaded dark green.



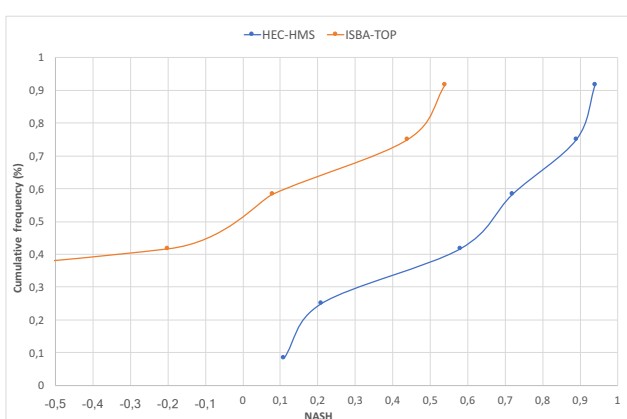

**Figure 3.** Cumulated Nash efficiency frequencies for hourly discharges simulated with HEC-HMS (blue line) and ISBA-TOP (orange line) within $REF2011$ and $REF2012$ experiments.





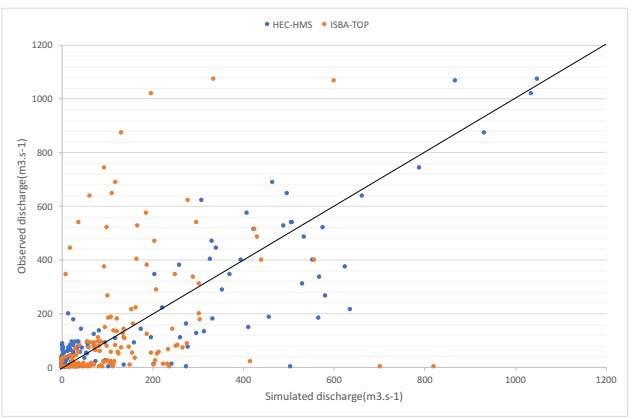

**Figure 4.** Hourly observed discharges versus simulated discharges ($m^3.s^{-1}$) for all watersheds simulated with HEC-HMS (blue points) and ISBA-TOP(orange points) within $REF2011$ and $REF2012$ experiments.

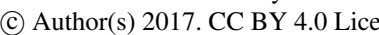



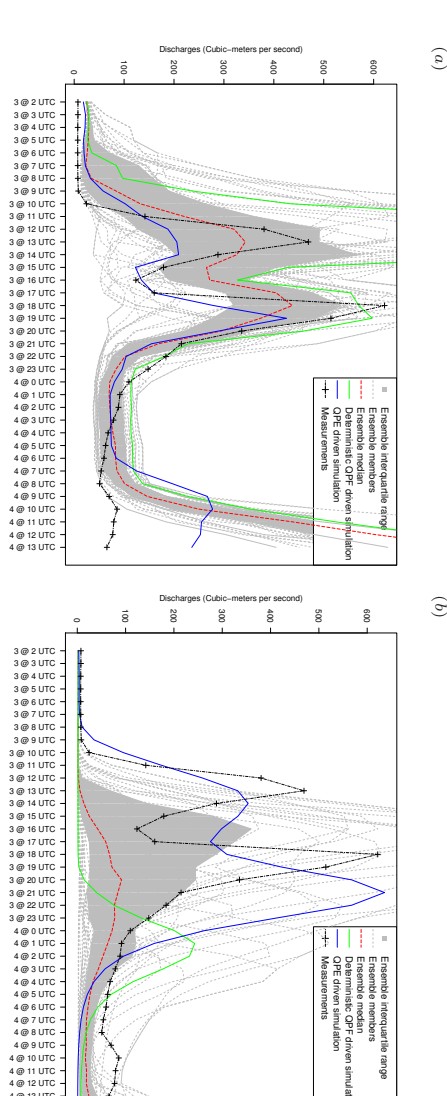

**Figure 5.** Simulated and observed hourly discharges of the Llobregat River at Castellbell for (a) the AROME *RUN03* experiments driving ISBA TOP, and (b) WRF *RUN03* experiments driving HEC-HMS. Black crosses denote the measurements from 03/11/2011 00UTC to 04/11/2011 12UTC. The blue lines the QPE driven discharge simulations. The green lines are the deterministic driven discharge forecasts. The grey dashed lines denote the individual HEPS members. The shaded areas represent the HEPS interquartile range. The red lines stand for the HEPS median.





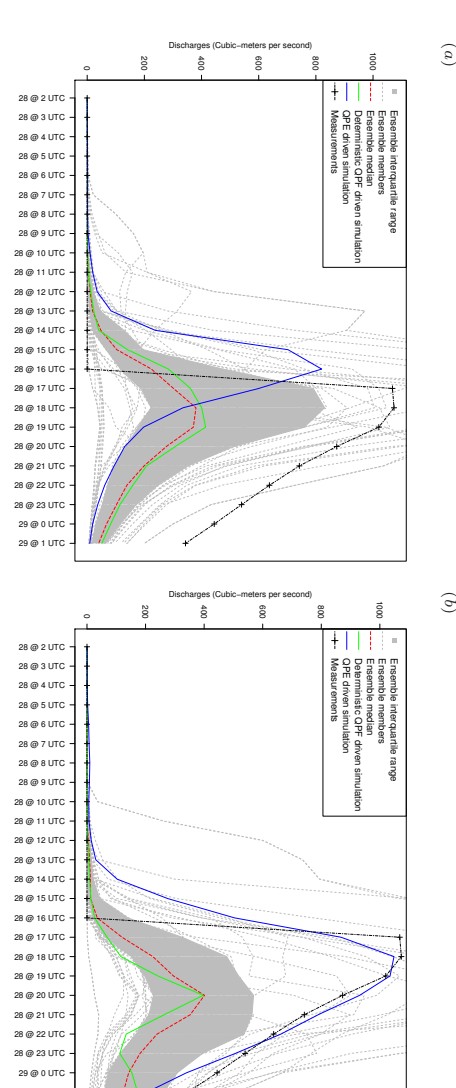

**Figure 6.** As Fig. 5 but for the Guadalentín river at Paretón and (a) the AROME RUN28 experiments driving ISBA-TOP, and (b) the WRF RUN28 experiments driving HEC-HMS from 28/09/2012 00UTC to 29/09/2012 00UTC.





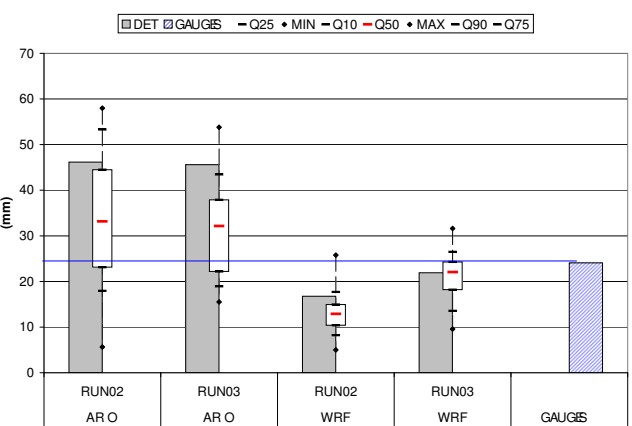

**Figure 7.** 24h-accumulated rainfall averaged on the Llobregat catchment starting on 03/11/2011 00UTC from raingauges data (purple column), deterministic QPF (shaded columns) and ensemble forecasts (boxplots) from AROME and WRF 48-h runs started on 02/11/2011 at 00UTC ($RUN02$) or 03/11/2011 at 00UTC ($RUN03$).





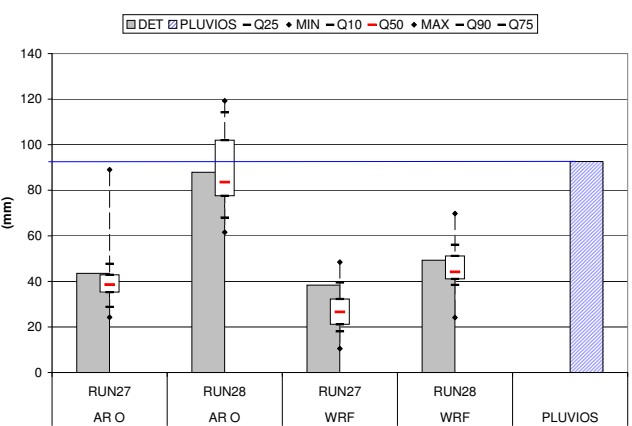

**Figure 8.** 24h-accumulated rainfall averaged on the Salabosque catchment starting on 28/09/2012 00UTC from raingauges data (purple column), deterministic and ensemble forecasts from AROME and WRF 48-h runs started on 27/09/2012 00UTC ($RUN27$) or 28/09/2012 00UTC ($RUN28$).





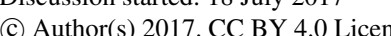

**Figure 9.** 24-h accumulated precipitation (03/11/2011 00UTC - 04/11/2011 00UTC) from raingauges (a), $90^{th}$-percentile of QPF from AROME ensemble RUN02 (b), RUN03 (c) and WRF ensemble RUN02 (d) and RUN03 (e) over the Llobregat catchment.





**Figure 10.** 24-h accumulated precipitation (28/09/2012 00UTC - 29/09/2012 00UTC) from raingauges (a), $90^{th}$-percentile of QPF from AROME ensemble RUN27 (b), RUN28 (c) and WRF ensemble RUN27 (d) and RUN28 (e) over the Guadalentín catchment.




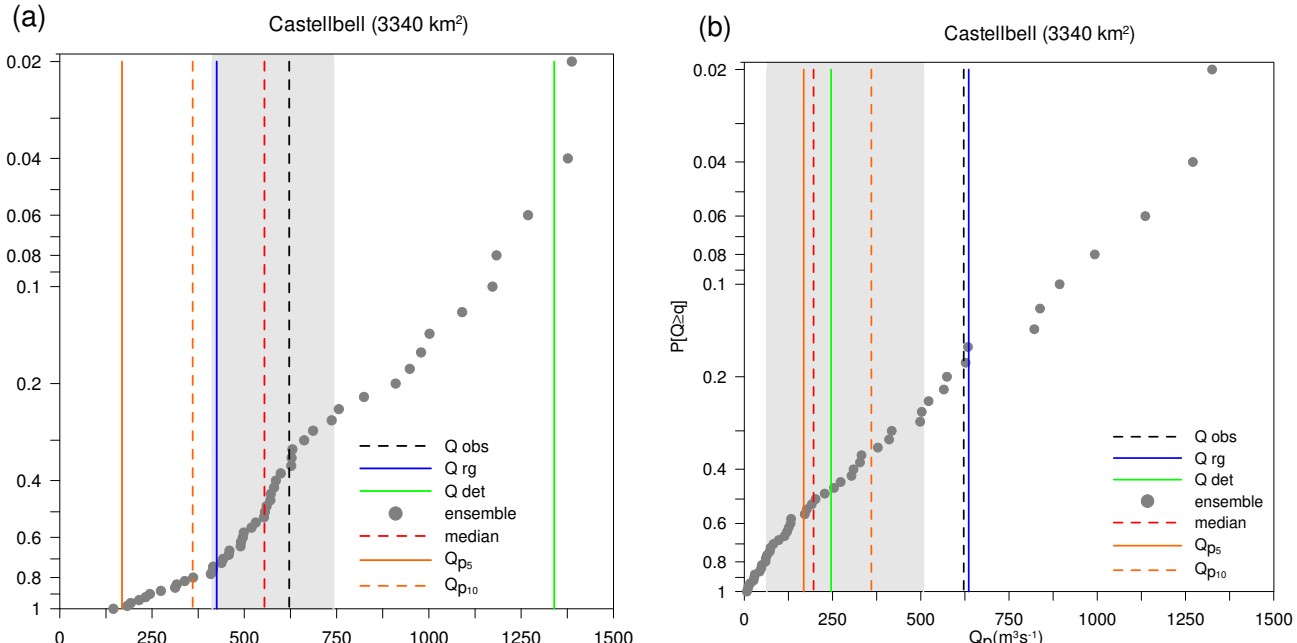

**Figure 11.** Peak discharge exceedance probabilities for the 02 November 2011 hydrometeorological episode at Castellbell for (a) AROME driven ISBA-TOP and (b) WRF driven HEC-HMS runoff experiments. The vertical dashed black line denotes the observed maximum flow. The solid blue and green lines correspond to the rain-gauge and NWP-deterministic driven maximum discharges, respectively. The dash red line represents the ensemble median peak discharge. The light gray shaded area depicts the ensemble spread between quantiles q0.25 and q0.75 of the members. Variables $Qp_5$ and $Qp_{10}$ denote peak discharge exceedance probabilities of the 5-, 10-year return periods, respectively.





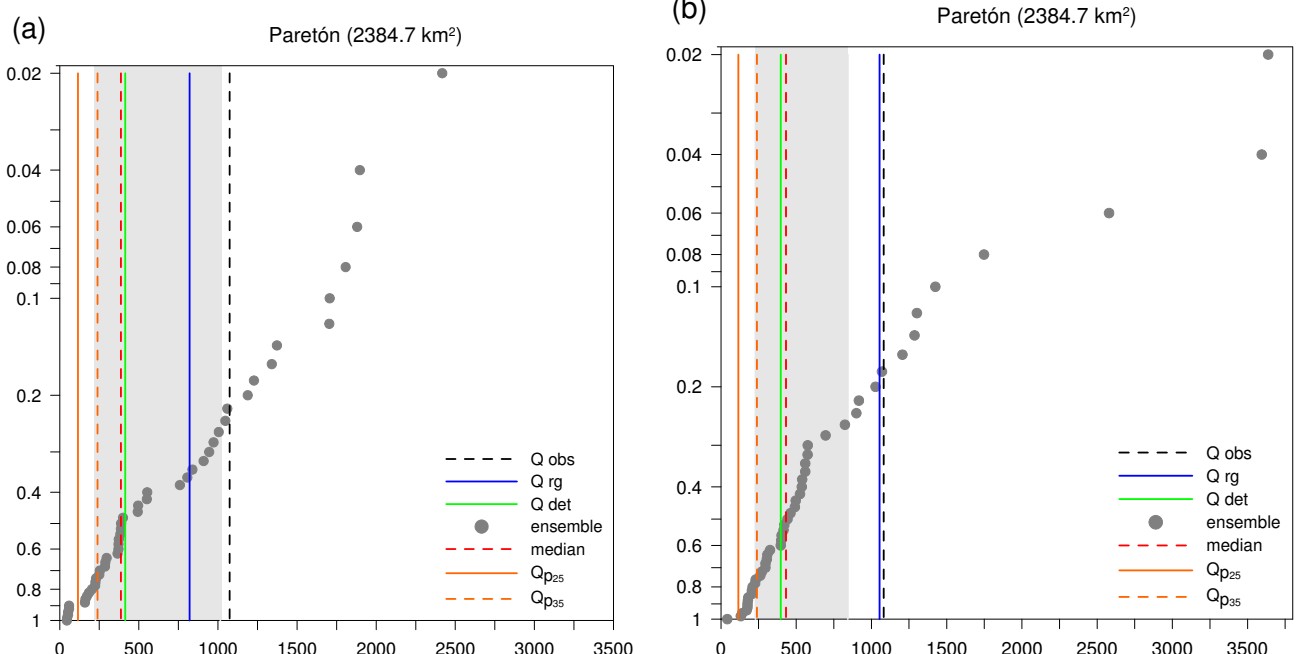

**Figure 12.** As figure 11 but for the 28 September 2012 hydrometeorological episode at Paretón. Variables $Qp_{25}$ and $Qp_{35}$ denote peak discharge exceedance probabilities of the 25- and 35-year return periods, respectively.




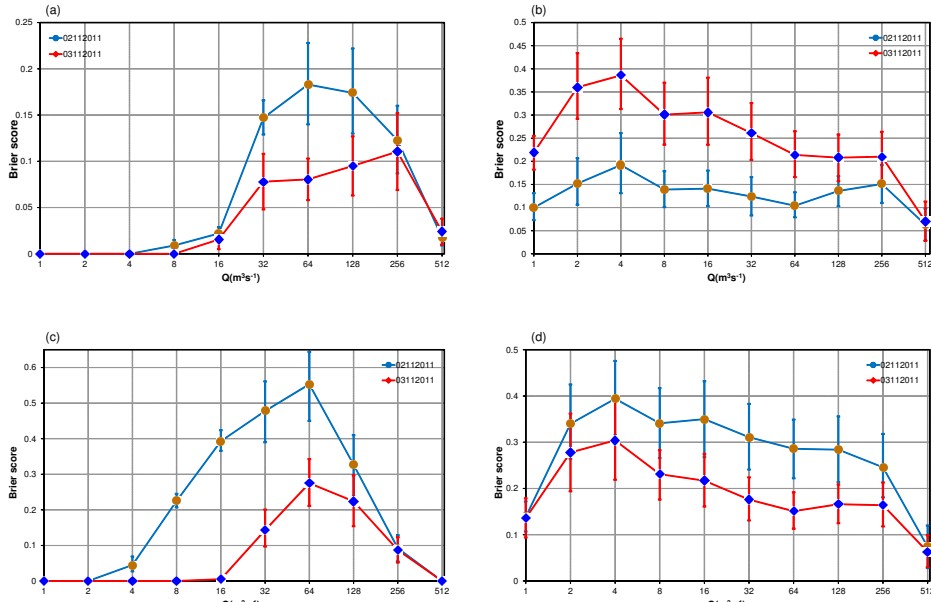

**Figure 13.** Brier Scores of discharges forecasts by AROME-ISBA ensemble (a), AROME-HMS ensemble (b), WRF-ISBA ensemble (c) and WRF-HMS ensemble (d) for the November 2011 study case . The reference is raingauge driven discharges simulation with ISBA-TOP for a and c and with WRF for b and d. The blue line is obtained for $RUN02$ experiments, the red line is obtained for $RUN03$ experiments.



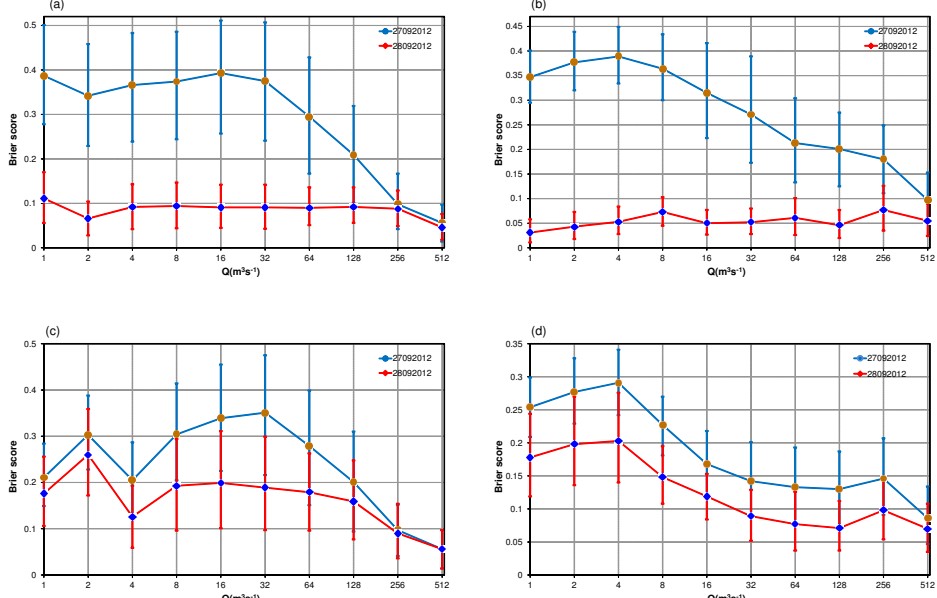

**Figure 14.** As Fig. 13 but for the September 2012 study case. The blue line is obtained for $RUN27$ experiments, the red line is obtained for $RUN28$ experiments.




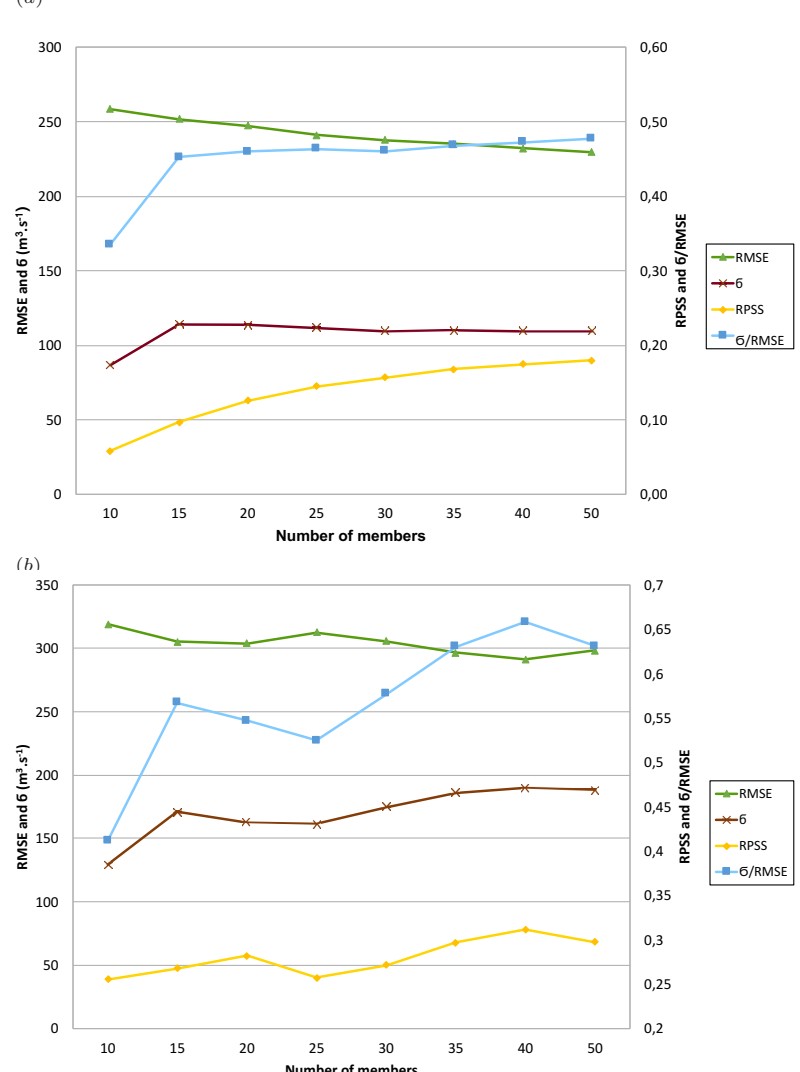

**Figure 15.** Probabilistic Scores of discharges forecasts by AROME-ISBA (a) and WRF-HMS (b) for the September 2012 study case. The reference is discharges simulations obtained with each hydrological model driven by rainfal observations.





**Table 1.** Summary of all the numerical experiments performed for the 03 November 2011 and 28 September 2012 flash floods.

| Name of the experiment | Starting time | Meteorological data | Hydrological models | Catchments where discharge is simulated | Number of simulated discharge time-series |
|---|---|---|---|---|---|
| **November 2011 event** | | | | | |
| REF2011 | | QPE | HEC-HMS | Lorca/Paretón/Salabosque | 3 |
| | | " | ISBA-TOP | " | 3 |
| RUN02 | 02/11/2011 at 00UTC | AROME deterministic QPF | HEC-HMS | " | 3 |
| | " | " | ISBA-TOP | " | 3 |
| | " | AROME ensemble QPF | HEC-HMS | " | 150 |
| | " | " | ISBA-TOP | " | 150 |
| | " | WRF deterministic QPF | HEC-HMS | " | 3 |
| | " | " | ISBA-TOP | " | 3 |
| | " | WRF ensemble QPF | HEC-HMS | " | 150 |
| | " | " | ISBA-TOP | " | 150 |
| RUN03 | 03/11/2011 at 00UTC | AROME deterministic QPF | HEC-HMS | " | 3 |
| | " | " | ISBA-TOP | " | 3 |
| | " | AROME ensemble QPF | HEC-HMS | " | 150 |
| | " | " | ISBA-TOP | " | 150 |
| | " | WRF deterministic QPF | HEC-HMS | " | 3 |
| | " | " | ISBA-TOP | " | 3 |
| | " | WRF ensemble QPF | HEC-HMS | " | 150 |
| | " | " | ISBA-TOP | " | 150 |
| **September 2012 event** | | | | | |
| REF2012 | | QPE | HEC-HMS | Castelbell/Sadurní/Despí | 3 |
| | | " | ISBA-TOP | " | 3 |
| RUN27 | 27/09/2012 at 00UTC | AROME deterministic QPF | HEC-HMS | " | 3 |
| | " | " | ISBA-TOP | " | 3 |
| | " | AROME ensemble QPF | HEC-HMS | " | 150 |
| | " | " | ISBA-TOP | " | 150 |
| | " | WRF deterministic QPF | HEC-HMS | " | 3 |
| | " | " | ISBA-TOP | " | 3 |
| | " | WRF ensemble QPF | HEC-HMS | " | 150 |
| | " | " | ISBA-TOP | " | 150 |
| RUN28 | 28/09/2012 at 00UTC | AROME deterministic QPF | HEC-HMS | " | 3 |
| | " | " | ISBA-TOP | " | 3 |
| | " | AROME ensemble QPF | HEC-HMS | " | 150 |
| | " | " | ISBA-TOP | " | 150 |
| | " | WRF deterministic QPF | HEC-HMS | " | 3 |
| | " | " | ISBA-TOP | " | 3 |
| | " | WRF ensemble QPF | HEC-HMS | " | 150 |
| | " | " | ISBA-TOP | " | 150 |





**Table 2.** Scores for hourly discharges ($m^3.s^{-1}$) with the meteorological ensembles AROME-EPS and WRF-EPS and the hydrological models ISBA-TOP and HEC-HMS. All catchments and cases are considered.

| Meteorological model | AROME | WRF | AROME | WRF |
| --- | --- | --- | --- | --- |
| Hydrological model | ISBA-TOP | ISBA-TOP | HEC-HMS | HEC-HMS |
| $RPSS$ | 0.41 | 0.02 | 0.46 | 0.13 |
| $RMSE\ (m^3.s^{-1})$ | 222.1 | 233.3 | 229.9 | 219.6 |
| $\sigma\ (m^3.s^{-1})$ | 73.4 | 23.9 | 220.5 | 50.2 |
| $\frac{\sigma}{RMSE}$ | 0.33 | 0.10 | 0.95 | 0.23 |





**Table 3.** Peak discharge exceedance probabilities $[P(Q \geq q)]$ for the AROME-ISBA and WRF-HEC HEPSs for the 03 November 2011 and 28 September 2012 episodes and the indicated stream gauges. Variables $Qp_5$, $Qp_{10}$, $Qp_{25}$ and $Qp_{35}$ denote peak discharge exceedance probabilities of the 5-, 10-, 25- and 35-year return periods, respectively.

| Episode | 03/11/2011 | 28/09/2012 |
|---|---|---|
| Streamgauge | Castellbell | Paretón |
| Experiment | $[P(Q \geq Qp_{obs} = 621.8m^3.s^{-1})]$ | $[P(Q \geq Qp_{obs} = 1073.3m^3.s^{-1})]$ |
| AROME-ISBA | 0.36 | 0.20 |
| WRF-HEC | 0.18 | 0.16 |
| Experiment | $[P(Q \geq Qp_5 = 168.5m^3.s^{-1})]$ | $[P(Q \geq Qp_{25} = 115.0m^3.s^{-1})]$ |
| AROME-ISBA | 0.98 | 0.88 |
| WRF-HEC | 0.56 | 0.98 |
| Experiment | $[P(Q \geq Qp_{10} = 359.9m^3.s^{-1})]$ | $[P(Q \geq Qp_{35} = 238.5m^3.s^{-1})]$ |
| AROME-ISBA | 0.80 | 0.72 |
| WRF-HEC | 0.34 | 0.74 |