# Peer review of "Flash-flood forecasting in two Spanish Mediterranean catchments: a comparison of distinct hydrometeorological ensemble prediction strategies"

_Hydrology and Earth System Sciences, 2017_

## Referee Comment (RC1) · Anonymous Referee #1 · 17 Aug 2017

Overview

The manuscript describes the application of two Short-Range Ensemble Prediction Systems (SREPS) for the forecast of two heavy precipitation events that affected two different semi-arid Mediterranean catchments in Spain. The forecast performance has been evaluated in terms of the accuracy of the quantitative precipitation forecasts, using also two hydrological models for flood simulations as additional verification tool. The two hydrological models are characterized by different structures and physical parameterizations. In my opinion this manuscript does not show features to be consid-

ered as a cutting-edge case study that deserves to be published in HESS. The present manuscript does not introduce an innovative use of ensembles in hydrological forecasts and provide original recommendations about the interpretation of outcomes from meteo-hydrological model coupling. Actually, the considered meteorological and hydrological models were already tested in several previous studies, some of them also from the same authors. Furthermore, one of the two case studies was already investigated in a past work by one of authors, testing a near identical meteo-hydrological forecasting chain. A real multi-model approach is not proposed (for instance, a post-processing approach that merges the input from both meteorological ensembles or the outcomes from both hydrological models would deserve to be considered); just a separate evaluation (and comparison) of outcomes from the coupling of each pair of meteo-hydrological model is discussed. The large majority of conclusions are quite expected and duplicates previous studies of both authors. Finally, the study suffers from some potential fatal flaws about the calibration/validation of the two hydrological models and the application of the method to generate one of the ensemble forecasts (see general comments for more details), that may affect the interpretation of results. Summarizing, despite of the efforts of the authors and the type of the manuscript (i.e., cutting-edge case studies), I deem that this manuscript fails to meet the requirements to be published in HESS.

General comments

(1) The SREPS approach should be deeper introduced and described, acknowledging specific references on the subject over the last years.

(2) One of the two case studies (i.e., the 2012 event) was already investigated in Amengual et al. (2015), testing a near identical meteo-hydrological forecasting chain. Actually, the version of WRF used in this study differs slightly from the version of WRF used in the previous study (in particular, for the higher horizontal resolution and number of vertical levels). However, no discussion about this issue is provided in the present manuscript. Authors should comment the impact of these upgrades (for instance, by

showing and discussing whether the rainfall forecast accuracy for hydrological applications improves with the latest version of WRF, preferably testing a longer period or more case studies), in order to determine the model sensitivity to the atmospheric processes leading to the high precipitation amounts.

(3) The two meteorological models have the same nominal horizontal resolution. However, the ensemble based on AROME-WMED may be characterized by an actual horizontal resolution that is higher than the ensemble based on WRF, due to the different approaches used to generate the ensemble (that is, the AROME-WMED approach inherently contains a sort of downscaling due to the error climatology correction). This issue may be an additional subject to be investigated in order to improve the contents of the manuscript.

(4) The description of the application of the perturbation method to generate an ensemble forecasts starting from the deterministic AROME-WMED forecast lacks of some details. Based on the actual manuscript, I guess that this approach was not properly adjusted to the investigated study areas (if right, this guesswork could partly justify the poorer performance provided by the AROME-WMED based ensemble with respect to the WRF-based ensemble). It seems that authors have used the statistics of the perturbation method estimated by Vincendon et al. (2011) over a different area and period for the implementation of the method over the two case studies of the present manuscript.

(5) The calibration/validation task used for the two hydrological models may hamper a proper comparison of performance. On the one hand, HEC-HMS is calibrated by considering (at least) one of the investigated case studies (if inferred correctly from Amengual et al., 2015). On the other hand, ISBA-TOP did not require a specific calibration for the selected study catchments (as stated at Pag9, line 19). The potential overfitting of the HEC-HMS parameters could influence the comparison on the impact of different model structures (which is one of aims of the manuscript, as stated at Pag2, lines 22-34). Furthermore, the calibration performed considering the investigated case

studies may not conciliate with a simulation of real-time application of the forecasting chain driven by HEC-HMS (which is one of aims of the manuscript, as stated at Pag2, lines 25-26).

(6) A detailed analysis about the role of each model characteristic on determining the discharge output is not discussed throughout the manuscript. Just an overall evaluation of each pair of meteo-hydrological model coupling is provided in Sections 4 and 5, with some additional comments about the comparison of performances. A multi-model approach is not really proposed. For instance, a statistical post-processing approach that merges the input from both meteorological ensembles (to be then used as a single input to hydrological simulations) or the outcomes from both hydrological models (driven by the same input) would deserve to be considered as an original contribution and added value for the manuscript. The results of such a merging could then be compared to outcomes of the single meteo-hydrological forecasting chains.

(7) The conclusions are quite expected or similar to conclusions in Amengual et al. (2015) and Vincendon et al. (2011), for instance statements at Pag15, lines 11-13, 16-19 and 22-24.

Specific comments

Pag1, lines 13-14: Please specify that HEC-HMS is a semi-distributed and conceptual-based model and ISBA-TOP a fully-distributed and physically-based model (as done at pag2, lines 19-21).

Pag1, lines 14-15: This summary of results does not highlight new and original outcomes.

Pag2, line 15: The date of the event in November 2011 seems to be outside the period described at pag2, lines 2-5.

Pag2, lines 27-34: A detailed analysis about the role of each model characteristic on determining the discharge output is not discussed throughout the manuscript. Just

an overall evaluation of each pair of meteo-hydrological model coupling is provided in Sections 4 and 5.

Pag3, lines 2-4: This sentence is questionable and contrasts with the sentence at pag3, lines 8-10. Could authors add references to prove this statement?

Pag3, lines 4-5: This sentence is not clear. Would author refer to the fact that nowadays the high resolution of NWP models matches the resolution of finer distributed hydrological models? Please clarify.

Pag3, lines 11-13: SREPSs should be deeper introduced and described, using specific references.

Pag4, lines 12-14: Four stations are cited, but results are discussed just for three out of four stations in the following sections.

Pag5, lines 5-6: Three stations are cited, but results are discussed just for two out of three stations in the following sections.

Pag7, lines 1-2: "AROME QPF". Do authors refer to QPF provided by AROME-France? Please clarify.

Pag7, lines 10-21: Which data have been used to perform the analysis here described? Was the error climatology estimated using data from the two case studies of the manuscript or from a longer database? Please add details about this issue. Furthermore, more details should be added for the description of the perturbation method. How is the lead time taken into account in the implementation of the perturbation method?

Pag7, lines 22-23: This sentence is not clear. Why is it necessary that the statistics of AROME-WMED and AROME-France are close to apply the perturbation method described at lines 17-21? A reader could guess that authors have used the statistics of the perturbation method estimated by Vincendon et al. (2011) over a different area and period for the implementation of the method over the two case studies of the present

manuscript.

Pag8, line 1: Please specify to which grid authors are referring and its features (i.e., dimensions of the grid cell).

Pag8, lines 24-25: Please add details about the calibration and validation periods. Were the two case studies of the present manuscript used to calibrate/validate the set-up of HEC-HMS used in this study? From Amengual et al. (2015) it seems that at least the 2012 event was used to calibrate HEC-HMS. So, on the one hand, HEC-HMS is calibrated over (at least) one of the investigated case studies; on the other hand, ISBA-TOP does not require a calibration (as stated at pag.9, line 19). If this is right, the potential overfitting of the HEC-HMS parameters could hamper a proper evaluation of the comparison on the impact of different model structures (which is one of aims of the manuscript, as stated at Pag2, lines 22-34). Furthermore, the calibration/validation task performed considering the investigated case studies may not conciliate with a simulation of real-time application of the forecasting chain driven by HEC-HMS (which is one of aims of the manuscript, as stated at Pag2, lines 25-26).

Pag8, lines 29-30: Which is the added value of using a 5 minute time-step for the simulation of HEC-HMS? May such a spatial-temporal detail of the rainfall input be likely dampened by the semi-distributed feature of HEC-HMS? On the one hand, observed and forecast rainfall data were originally provided at a 1 hour time-step and a linear interpolation to fit rainfall for shorter periods may be questionable and not significant (with respect to a real temporal evolution of the rainfall event). On the other hand, ISBA-TOP was run with an hourly time-step.

Pag9, line 4: Please specify the dimension of the grid cell.

Pag10, lines 19-23: Five river section are here cited; but, six sections were considered in Table 1 and seven sections were mentioned in the description of catchments in Section 2. This may be confusing. Please consider to mention throughout the paper just the five sections for which results are discussed.
[Figure]

Pag10, line 24: "all outlets". Do authors refer to the five sections cited at pag.10, lines 21-22?

Pag10, line 26: Please add details about the method of Le Lay and Saulnier (2007).

Pag10, line 30: Were the Nash efficiencies computed over hourly outputs of both hydrological models? Or were 5-minute outputs considered for HEC-HMS? Please clarify.

Pag11, line 1: The contents of Fig.4 are not meaningful. The words "predictive uncertainty" are not properly used and could be misleading with respect to the treating of the uncertainties within the hydrological forecasting context (see, for instance, Krzysztofowicz, 1999 and Todini, 2008). It is quite simplistic to infer the predictive uncertainty of both models by the scatter plot shown in Fig.4.

Pag11, lines 7-8: This statement is quite expected.

Pag11, lines 11-12: This statement is quite expected.

Pag11, lines 13-21: Please add warning levels in the figures 5 and 6, in case of these values are defined (for the aims of public safety). These reference values may allow to better evaluate the error magnitude, especially with respect to the ensemble spread.

Pag11, lines 19-21: Please mention the river section to which refer the discussed results (shown in Fig.6). Are these outcomes still valid also for the other sections of the same river?

Pag11, lines 22-24: Authors should try to deeper relate these outcomes with the different structures of the two hydrological models.

Pag11, line 24: By a real-time point of view, a multi-model approach can be really informative to end users (forecasters as well as authorities in charge of decisions in case of flood) if the "climatology" of the response of each hydrological model to a certain input is well "known", based on a long dataset of simulations. Otherwise, the mixing of outputs from different forecasting chains could increase the forecast uncertainty

and the difficulties to interpret such outcomes. For instance, have authors evaluated if the better accuracy of ISBA-TOP to simulate the dynamic processes (or HEC-HMS to simulate the peak amplitude) is systematic?

Pag11, line 30: Please specify which is the closure section of the basin that is considered to compute the catchment-averaged precipitation amounts.

Pag11, line 30: The extent of the overestimation is quite larger (near twice the observed amount) with respect to the extent of the underestimation. This outcome should be highlighted.

Pag12, line 1: The spread of AROME is very high for RUN02, decreasing its reliability and making the forecast near worthless by a operational point of view. Could authors justify such a outcome?

Pag12, line 3: Please specify which is the closure section of the basin that is considered to compute the 24-h accumulated rainfall averages.

Pag12, lines 5-7: This sentence is quite generic. Moreover, it confirms that a multi-model approach could be misleading and worthless if the error climatology of models is not properly known over the study area.

Pag12, lines 13-19: The perturbation method seems to be not able to correct the errors of the deterministic run for this case study.

Pag12, lines 19-22: It seems that the proposed methodologies to built rainfall ensembles are not the best solutions for the investigated study areas.

Pag12, lines 28-30: It is quite expected, or it is likely, that the a forecast interval provided by an ensemble better encompass a reference simulation than a deterministic forecast. Authors should also comment the performance of the ensemble median.

Pag12, lines 32-34: Authors should deeper investigate such a strange outcome.

Pag13, lines 1-2: Why are the results just shown (in Fig. 5, 6, 11, 12 and Table 3)

for these two sections? Are the outcomes representative also for the remaining river sections? Please clarify.

Pag13, line 6: Authors cite a value of exceedance probability (i.e., 0.2) that is used by AEMET to issue warning for extreme weather events. How can this value be transferred to evaluate the "extremeness" of streamflows?

Pag13, lines 6-9: Why are different return periods used to quantify the risk of flood for the two sections (Qp5 versus Qp25 and Qp10 versus Qp35)?

Pag13, line 10: Please specify the range of exceedance probabilities that characterize the low and moderate classes.

Pag13, line 16: "all the outlets". Do authors refer to the five sections cited at pag.10, lines 21-22? Please clarify.

Pag13, lines 22-23: It is quite expected (by construction) that the RPS rewards ensemble forecast when compared to deterministic forecasts. The values in Table 2 seems to contrast the contents of Fig.15. "RPSS is lower". The improvement is near negligible.

Pag13, lines 27-30: Have authors verified this statement by analyzing the synoptic and large mesoscale dynamical and thermodynamical environment of the ensemble members?

Pag13, lines 30-34: Otherwise, the data used to estimate the climatology of errors are weakly related to the investigated study areas, so that the variability (uncertainty) increases (larger spread).

Pag14, lines 2-3: Were the Brier scores computed using hourly discharge values? How was the dimension of bins for the computation of BS scores chosen? How were values greater than 512 m3/s treated in the computation of BS (were they assigned to the latest binning?)? Please clarify and specify to which river sections refer the results shown in Fig. 13 and 14.

Pag14, lines 3-4: Please add details about the implementation of bootstrap.

Pag14, line 21: A bootstrap approach could be applied also for the test on the ensemble size.

Pag14, line 23: Please specify to which river sections refer the results shown in Fig. 15.

Pag14, lines 24-26: Maybe, this outcome is related to the method used to build the ensemble (the AROME-based ensemble shows lower variability).

Pag14, lines 27-31: This statement confirms that the present study does not highlight new or meaningful outcomes and not provide useful recommendations.

Pag15, lines 9-10: It is quite expected that a distributed model better reproduces the flood dynamics with respect to a semi-distributed model. Was HEC-HMS calibrated considering also the two investigated case studies? If yes, it is quite expected that HEC-HMS better simulate the peaks with respect to the uncalibrated ISBA-TOP.

Pag15, lines 14-15: It is not clear if the higher spread of AROME is considered as a valuable characteristics or a drawback. Please clarify the meaning of this sentence.

Pag15, lines 21-22: This sentence is not clear. Have the tested ensembles different maximum lead-times?

Pag15, lines 29-32: It is not showed that it is convenient to use a multi-model and ensemble approach, given that the statistical analyses are independently carried out and discussed for each pair of model coupling. A combined approach that merges the outcomes from both hydrological models (driven by the same input) or from both meteorological ensembles (for instance, different percentiles taken from the whole members of both ensembles) to be used as single input to hydrological simulations was not considered and discussed. Such a comparison should deserve to be discussed.

Technical corrections

Pag2, line 4: "Mediterranéean" should be "Mediterranean".

Pag2, line 23: "allow" should be "allows".

Pag3, line 18: Should "04" be "03"?

Pag3, line 27: "overviiew" should be "overview".

Pag3, line 27: "flsh-floods" should be "flash-floods".

Pag7, line 15: Should "tunned" be "tuned"?

Pag7, line 33: Should "applysing" be "applying"?

Pag10, line 3: "liste" should be "listed".

Pag11, line 15: Should "Despì and" be "Despì (not shown) and"?

Pag11, line 16: Should "Sadurni." be "Sadurni (not shown)."?

Pag11, line 17: Should "magnitude (Fig. 5b)" be "magnitude at Castellbell (Fig. 5b)"?

Pag11, line 17: Should "is obtained." be "is obtained (not shown)."?

Pag11, line 31: "QPFs QPF" should be "QPFs".

Pag12, line 1: Should "scenarii" be "scenarios"?

Pag12, line 4: Should "scenarii" be "scenarios"?

Pag12, line 4: Should "suceed" be "succeed"?

Pag12, line 12: Should "overestimated" be "overestimate"?

Pag12, line 13: Please consider to replace "biases" with "errors".

Pag12, line 14: Should "succed" be "succeed"?

Pag12, line 32: Should "75%-percentile" be "median"?

Pag14, line 23: Should "range from 10 to 50 members" be "ranges from 10 to 40

members"?

Pag28: Should "Salabosque catchment" be "Guadalentin catchment" in the caption of Fig.8?

Pag31-32: In all the panels of Fig. 11 and 12, it seems that the dash red line is used to represent the ensemble median peak discharge as well as the peak discharge exceedance probabilities of the 10-year return period (Qp10). Please use a different label for one of the two values.

Pag33-34: In all the panels of Fig. 13 and 14, please consider to enlarge the text legend (i.e., date of the run) and remove the black grid lines of fixed x and y value.

Pag35: Please consider to use the same scale for the y-axis in both the panels of Fig. 15 (that is, maximum value of 350 in the left y-axis and range 0-0.7 in the right y-axis).

References used in the present review Krzysztofowicz, R. Bayesian theory of probabilistic forecasting. Water Resources Research, 35 (9), 2739–2750, 1999. Todini, E. 2008. A model conditional processor to assess predictive uncertainty in flood forecasting. Intl. J. River Basin Management, 6(2): 123-137.

---

## Referee Comment (RC2) · Anonymous Referee #2 · 18 Aug 2017

The present manuscript describes a comparison of different HEPSs on two Spanish catchments on two flash floods events. The HEPSs are based on two meteorological models configurations and two hydrological models, and are assessed with a set of numerical tools.

While the topic is of interest for the journal, the manuscript is far from what can be published in it in my opinion. Indeed, it suffers from presentational flaws, lack of adequate reference to past works, lack of background on this research topic, lack of deep analysis, lack of innovative methods or results, poor English, many technical issues. I

strongly agree with reviewer 1 and I think that the paper should be rejected.

Moreover, the manuscript type does not correspond to a "cutting-edge case studies report" as I understand it. It does not "broaden the knowledge base in hydrology" and data or models are not shared (as far as I know).

Please find below my major and minor remarks. Please note that due to the high number of technical mistakes, and because many parts of the manuscript should be rewritten, I did not list all of these technical mistakes. I believe that most of these minor technical mistakes should have been dealt with by authors before submission

Major remarks

- The English is sometimes deficient; many mistakes are present in the manuscript. Correction by a native speaker should be made

- The introduction provides profusion of references from the two authors (although it is not always used for highlighting the challenges or methodologies, as said by reviewer 1). However, very few is given about initiative such as HEPEX (that is the HEPS working group initiative, that may be of interest I think). Comparisons of forecasts from different strategies are not new and some of them should be acknowledged. Some works from the first authors lab compared short-range hydrological ensemble forecasts using two EPSs with a relatively well-defined framework (Thirel et al., 2008) or two hydrological models (Randrianasolo et al., 2010): why not referring to that? It does use ISBA and ECMWF EPS for instance!

- P. 9, L. 23: those are rather variables or inputs than parameters. In my opinion, it is not a good idea to use a complex fully-distributed physically-based model, supposed to better represent the processes leading to flash floods, if you put in it constant values for very physical inputs that have an impact on the rainfall-runoff relationship. Also, we have no idea about the constant values that were chosen.

- The scores used in the manuscript should be properly defined and use literature

references either in the methodology section or in an Appendix. Due to the many approximation present in the manuscript, it may help clarifying what is really assessed

- Section 4.1 title and implied goal is misleading in my opinion: the authors are not assessing or discussing the hydrological models uncertainty (there is no hydrological model calibration / validation or sensitivity analyses in the manuscript). They are showing what the impact of the two hydrological models is on the flood simulation. That is very different.

- Figure 3 is unclear: do we have here both references and all gauges mixed?

- Figure 4 is not about predictive uncertainty

- P. 11, L. 6-9: I am a bit puzzled by these sentences. Of course if you do not calibrate your physically-based model, you can expect poorer results... But that is your choice! You cannot blame anyone for this! Even so-called physically-based models are full of approximations that make them to some degrees conceptual models (see Hrachowitz and Clark, 2017). And in the same time, of course conceptual models do not give good performances if they are not calibrated... They are made for being used after calibration!

- Section 4.2.3: from such a section title, one expects more than just comparing forecasts issued at two different dates before the event. We need a longer archive for this kind of impact study. Conclusions are likely to be just opposite on many other events.

- L. 21: do you mean you randomly selected n member out of 50 once each time, or did you randomly do that many times to account for the sampling effect (i.e. the —non—selection of (a) specific member(s) can have a large impact on the scores)?

- Section 4.2.4: I am a bit puzzled by this section: if you use models as complex as ISBA, why would you discard some forecasts? You can easily miss useful information about extremes of the forecasts! Moreover, as all members are equi-probable, you modify the pdf of the forecasts by selecting some members, which is not correct

Minor remarks

- No space before colons

- P. 2, L. 6: near what?

- L. 7: what is IOPS?

- L. 8-9: a sentence usually contains a subject, a verb, and if necessary a complement

- L. 15: what is a level of HEPS?

- P. 2, L. 4: no accent in Mediterranean

- L. 6: usually figures do not need to be cited in introductions, more especially in the first paragraph

- L. 16: "basins" is missing after Llobregat?

- L. 17: capital letter to River after the river name (check whole document)

- L. 25-26: I feel that the sequencing of words is erroneous in this sentence

- P. 3, L. 20: Centre, not Center

- L. 27: two words are misspelled

- L. 28: why talking about section 4.2 specifically? Isn't it section 4?

- P. 4, L. 10: data ARE

- L. 16: occurred is misspelled

- L. 17: I think that citations must have an alphabetical or chronological ordering

- L. 18 and 21: the date format must not differ

- L. 33: altitude instead of height?

- P. 5, L. 4: stream-flow had a different writing in the manuscript up to now

[Figure]

- L. 22: the writing of "mm" must not differ, especially in the same line!

- L. 27: has instead of have

- P. 6, L. 14 and 28: the time format must not differ

- L. 23 and 24: these citations must be written with the following format: name et al. (year). Check whole document

- L. 28-29: months need capital letters in English

- P. 7, L. 3: rainfall should be singular

- L. 13-14: space or no space before mm? PDFs is misspelled

- L. 33: typo in applying

- P. 9, L. 26 and 31: space or no space between 00 and UTC?

- P. 10, L. 3: listed

- L. 24: the full name is Nash-Sutcliffe efficiency

- P. 11, L. 3: "Thus": what is the logical link with the previous sentence?

- P. 12, L. 4: succeed is misspelled

- L. 9: overestimateS

- L. 11: missing space before parenthesis

- L. 15: remove final point before parenthesis

- L. 16: reaches

- P. 13, L. 8: Table 3 is cited before Table 2!

- L. 15-21: most of it should belong to a methodology section

- L. 19: comparison is misspelled

- L. 22: no s to confirm

- L. 26: usually the sigma / RMSE ratio should be close to 1 over long time series. How true is it over single events? Please document!

- P. 14, L. 2: using BS instead of BSS here. You previously used RPSS! Moreover, you use all these scores as metrics, nothing else: no attempt is made to reflect on the impact of the statistical properties of the forecasts, this is very disappointing.

- L. 3: presented is misspelled

- L. 12: I would write "the later the simulations start"

- L. 13: "closely lead-times": is that grammatically correct?

- L. 21: Performance is misspelled, "is" should be place after the subject of the sentence.

- L. 24: for your information, works have been done to mathematically correct the impact of the number of members on some of the scores you use

- P. 15, L. 5: modelling had 2 "l" up to now in the manuscript

- L. 6: is it HEPSs? I think that "is" should be "are" here.

- L. 9: allows

- L. 11: "clearly" should be before "improves"

- L. 20: you have to choose between lead times and lead-times!

- L. 23-24: check there are two typos

- L. 25 "do", not "does"

- L. 27: what is a statistical study?

- L. 28: "responsible"?

- Figure 2: the stream gauges are very difficult to locate in these maps. The area value uses a different format than in P. 3, L. 32

- Figure 7-8: add "Rainfall" to the y-axis label. "Pluvios" is not correct in Fig. 8.

- Figure 11-12: these figures have panels of different sizes, and the x-axis label somehow sometimes appears incompletely

- Fig.13: in the caption, why are you talking about a reference? What is it used for?

- Fig. 15: please specify the unit of RPSS and sigma / RMSE. Rainfall is misspelled

- Table 2: please specify the unit of RPSS and sigma / RMSE

References:

Hrachowitz, M. and Clark, M. P.: HESS Opinions: The complementary merits of competing modelling philosophies in hydrology, Hydrol. Earth Syst. Sci., 21, 3953-3973, https://doi.org/10.5194/hess-21-3953-2017, 2017.

Randrianasolo, A., Ramos, M.H., Thirel, G., Andréassian, V., Martin, E. Comparing the scores of hydrological ensemble forecasts issued by two different hydrological models (2010) Atmospheric Science Letters, 11 (2), pp. 100-107.

Thirel, G., Rousset-Regimbeau, F., Martin, E., Habets, F. On the impact of short-range meteorological forecasts for ensemble stream flow predictions (2008) Journal of Hydrometeorology, 9 (6), pp. 1301-1317.

---

## Author Comment (AC1) · 22 Sep 2017

First, the authors would like to thank Reviewer #1 for its helpful comments. Unfortunately, we won't be able to address all the points within the time of the review. We have to withdraw the paper. But we would like to clarify some points. The first submission of this paper was not in the "cutting-edge case study" but the editor advised us to choose this option for submission. Of course the different models, meteorological or hydrological, have been used in other studies and papers. But we feel that it is useful to check if the results previously found on short samples remain valid on other cases and

catchments. We also find interesting to inter compare the hydrometeorological chains for the same case studies, this is favored by the Hymex context. We agree that a "post-processing approach that merges the input from both meteorological ensembles or the outcomes from both hydrological models would deserve to be considered". But it is difficult to achieve in the allotted time.

---

## Author Comment (AC2) · 22 Sep 2017

The authors also would like to thank Reviewer #2 for its helpful comments . As stated to Reviewer #1, we won't be able to address all the points within the time of the review and, after an initial submission of this paper, the editor advised us to propose it in "cutting-edge case study" section. All the data of this paper are available on simple request and are planned to be uploaded in the Hymex database. A correction of the paper by a native speaker cannot be achieved in the allotted time. We agree that the bibliography part could be improved. Thank you for the references you provided. The

paper will be withdrawn.